# Overexpression of *TpGSDMT* in Rice Seedlings Promotes High Levels of Glycine Betaine and Enhances Tolerance to Salt and Low Temperature

**DOI:** 10.3390/biom15111576

**Published:** 2025-11-10

**Authors:** Jinde Yu, Zihan Zhang, Ning Zhao, Xiaofei Feng, Dan Zong, Lihua Zhao

**Affiliations:** College of Biological Science and Food Engineering, Southwest Forestry University, Kunming 650224, China

**Keywords:** abiotic stress, glycine betaine, *Oryza sativa* L., overexpression, *TpGSDMT*

## Abstract

Salt and low temperature are serious abiotic stresses and important constraints to agricultural productivity across the globe. These abiotic stresses negatively affect plant growth and physiological, biochemical, and molecular processes. Glycine betaine (GB) is an important osmoprotectant that enables plants to resist salinity, low temperature, and drought. GB can be synthesized in many organisms, including animals, plants, and bacteria. In higher plants, GB is synthesized through two-step oxidation of choline. However, rice, an important food crop, cannot synthesize GB. Thus, conferring the ability to synthesize GB to rice through genetic engineering is of great significance for enhancing its tolerance to abiotic stress. Recently, an enzyme, GSDMT (glycine, sarcosine, and dimethylglycine methyltransferase) was found in a diatom, *Talassiosira pseudonana*, and found able to catalyze the three successive methylation steps of glycine to form GB. This biosynthetic pathway for GB synthesis is also the simplest in living organisms. Here, the optimized codon of the *TpGSDMT* gene sequence was synthesized and cloned into an overexpression vector, pBWA(V)HS, which contains a CaMV 35S promoter, and then, the constructed vector was transferred into rice (*Oryza sativa* L. ssp. *Japonica*). The GB content in transgenic rice showing overexpression of *TpGSDMT* was significantly increased, and these transformants exhibited markedly enhanced tolerance to salt and low temperature. These results indicate that the *TpGSDMT* gene can be used for the genetic improvement in crop plants’ resistance to salinity and low temperature.

## 1. Introduction

Abiotic stresses, such as salinity, low temperature, and drought, are important factors affecting crop growth, development, productivity, and quality [1,2,3,4]. When crops encounter these abiotic stresses, they can accumulate small molecular osmotic regulators, including glycine betaine (GB), polyols, and proline, in response to these stresses, thus enhancing the crop’s resistance to abiotic stresses [5,6,7,8,9,10]. Among these compatible solutes, GB is one of the most intensively studied osmoregulators, having numerous functions, including cellular osmotic adjustment, maintaining cell organelles and the integrity of the cell membrane, improving water use efficiency, and protecting various enzyme activities and the thermodynamic stability of proteins [11,12,13,14,15]. Meanwhile, GB indirectly activates the reactive oxygen species (ROS) scavenging system [7,16,17,18,19] and protects the structure and activity of the photosynthetic oxygen center [20,21,22]. Therefore, GB can improve the antioxidant capacity and reduce osmotic water loss, alleviating the inhibition of the photosynthetic capacity induced by abiotic stresses such as drought. Additionally, GB can also increase the germination rate of wheat seed and the biomass of their seedlings [23], as well as enhancing the development and size of tomato fruit [5,24].

GB is widely found in animals, plants, bacteria, and algae [17]. As of now, four GB biosynthesis pathways have been identified in living organisms, three of which are represented by the oxidation of choline and the other being the methylation of glycine [25,26,27]. In higher plants, the most well-known biosynthetic pathway of GB is the two-step oxidation of choline, namely, choline → betaine aldehyde → GB. The first and second steps are catalyzed by choline monooxygenase (CMO) and betaine aldehyde dehydrogenase (BADH), respectively [28]. In animals and some bacteria, the first step is catalyzed by choline dehydrogenase (CDH). The second-step enzyme is BADH, as in plants [29,30]. Some bacteria, such as *Arthrobacter globiformis*, directly convert choline into GB by a single enzyme, choline oxidase (COX), which is coded by the *CodA* gene, without BADH [31]. Another pathway of GB biosynthesis is the three-step methylation of glycine: glycine → sarcosine → dimethylglycine → GB. For example, in the halotolerant cyanobacterium *Aphanothece halophytica*, two N-methyltransferase enzymes, ApGSMT (glycine and sarcosine methyltransferase) and ApDMT (dimethylglycine methyltransferase), are involved in three methylation steps of GB synthesis: ApGSMT catalyzes the first two methylation steps of glycine to form sarcosine and dimethylglycine, respectively, and ApDMT catalyzes the methylation of dimethylglycine to form GB [32]. In an anaerobic phototrophic sulfur bacterium, *Ectothiorhodospira halochloris*, EcGSMT and EcSDMT (sarcosine and dimethylglycine methyltransferase) are involved in the three-step methylation of glycine: EcGSMT catalyzes the first two methylation steps and EcSDMT catalyzes the third methylation step [33]. In a halophilic methanoarchaeon, *Methanohalophilus portucalensis*, two methyltransferases, MpGSMT and MpSDMT, have been reported to be involved in the methylation of glycine [34,35]. Recently, a single gene, *TpGSDMT*, which codes methyltransferase (glycine, sarcosine, and dimethylglycine methyltransferase), was found to catalyze all three steps of the methylation of glycine to form GB, found in a diatom called *Talassiosira pseudonana CCMP1335* [27].

Although GB is widely found in living organisms, not all plants have the ability to accumulate GB [6]. For example, rice (*Oryza sativa*), tomato (*Lycopersicon esculentum*), tobacco (*Nicotiana tabacum*), and *Arabidopsis* are typically non-GB-accumulating species, even when these plants are subject to abiotic stress [36,37,38,39]. Previous studies have shown that these plants do not accumulate GB due to the function defect of the *CMO* gene or insufficient content of choline (the precursor of GB biosynthesis) [40,41]. Accordingly, a number of studies have used genetic engineering to transfer genes from GB-accumulating species such as spinach (*Spinacia oleracea*), *Arthrobacter globiformis* and *Aphanothece halophytica* into non-GB-accumulating plants such as rice, *Arabidopsis*, tobacco, and tomato [5,36,42,43]. These results showed that transgenic plants have the ability to synthesize GB and that transgenic plants showed enhanced resistance to abiotic stress and increased yield and biomass. However, there are also shortcomings in the above studies: Firstly, the GB content in transgenic plants is only 0.41–1.81 μmol g^−1^ FW (FW stands for fresh weight), which is much lower compared with plants that naturally synthesize GB, such as spinach, wheat (*Triticum aestivum*), and corn (*Zea mays*) (GB content is about 124–360 μmol g^−1^ FW in these plants) [40,41,43,44]. Secondly, the transgenic process is complicated because at least two genes for GB biosynthesis need to be transferred simultaneously into non-GB-accumulating plants [43]. Lastly, although the COX enzyme encoded by the *CodA* gene can directly oxidize choline to synthesize GB, insufficient choline content is one of the important reasons that prevent some plants from synthesizing GB [41]. Recently, Raldugina et al. directly delivered CodA protein into the tobacco plastid, the site of the substrate for CodA, and observed an increase in resistance in the transgenic tobacco at the level of primary and secondary biosynthesis of metabolites, alleviating the damage characteristic of salinity [45]. Meanwhile, exogenous GB application to wheat enhanced the antioxidative response and promoted the biosynthesis and accumulation of stress-responsive hormones such as abscisic acid (ABA) and salicylic acid (SA). Consequently, this treatment maintained better plant growth under drought stress and improved the root architecture systems and drought tolerance of wheat roots [46]. It has been found that GB can stimulate plant growth through a comprehensive improvement in photosynthesis and activate antioxidant enzymes such as superoxide dismutase (SOD) and catalase (CAT), thereby enhancing the salt stress response of tomato [47]. In addition, exogenous GB reduced Na toxicity in stevia (*Stevia rebaudiana*) by improving nitrogen metabolism, regulating polyamine metabolism, boosting antioxidant enzyme activities, protecting plasma membrane integrity, decreasing Na^+^ accumulation, and facilitating K^+^ uptake [48].

Rice, a non-GB-accumulating plant, is one of the most important food crops and feeds more than half of the world’s population [49,50,51], but it is highly sensitive to abiotic stress such as salinity, low temperature, and drought [52], and abiotic stress seriously affects crop productivity, growth, and development [53]. Given that GB plays a significant role in plant abiotic stress tolerance and can increase crop yield, it is important to improve the abiotic stress tolerance of rice by genetic engineering to make it synthesize GB simply and efficiently. Here, we transferred the *TpGSDMT* gene from *Talassiosira pseudonana*, which encodes the protein that performs all three steps of the methylation of glycine to produce GB, into rice. Our results showed that the transgenic rice could accumulate significant amounts of GB and had strong resistance to salt and low temperature. So, this study demonstrates a genetic engineering strategy for improving the stress resistance of non-GB-accumulating or poor-GB-accumulating plants and provides insights into the imbalance in sodium sensitivity and the disruption of vacuolar pH regulation caused by salt stress among different cell types [48]. Meanwhile, our research study can also provide excellent genetic resources for rice resistance breeding.

## 2. Materials and Methods

### 2.1. Plant Materials and Growth Conditions

Rice plants (*Oryza sativa* ssp. *japonica* cv. Nipponbare) were grown in a greenhouse under standard conditions (28/20 °C, day/night) with supplemental light (16 h photoperiod), unless specified otherwise. Primary transformed plants and transgenic progenies were planted in the greenhouse and transplanted into the field (Yuanjiang County, Yunnan, China) 60 days later. The planting density was 15 plants/m^2^, and during the field cultivation period, the average daily light exposure was 11.8 h, with a relative humidity of 67.3%.

### 2.2. Vector Construction of TpGSDMT and Transformation of Rice

The coding sequence (1725 bp) of the *TpGSDMT* gene (XM_002286728) was downloaded from the National Center for Biotechnology Information (NCBI), and the codons of *TpGSDMT* were optimized for expression in rice using the online tool ExpOptimizer (https://www.novopro.cn/tools/codon-optimization.html (accessed on 29 March 2023)). The optimized coding sequence (CDS) of *TpGSDMT* was subsequently synthesized at Majorbio Company (Shanghai, China). The synthesized CDS for *TpGSDMT* was amplified by PCR using the following primers: 5′-AACACGGGGGACTTTGCAACATGGCGCCTAATACCAGTACATCAACG-3′ and 5′-TGAAGACAGAGCTAGTTACATTAGCTGGATGGTGATTCGTCTACAACC-3′. The PCR product was inserted into a pBWA(V)HS vector, which contains elements including the 35S promoter, a hygromycin phosphotransferase selectable marker gene (*hyg*), and the nopaline synthase polyadenylation (nos) terminator, using the homologous recombination method (the vector was linearized by restriction endonuclease BsaI/Eco31I) [54]. The resultant construct, pBWA(V)HS-*TpGSDMT*, was transferred into *Agrobacterium tumefaciens* EHA105 and then transferred into rice by an *Agrobacterium*-mediated method [43]. The methodology for obtaining transgenic rice is as follows: Rice calli were infected with an *Agrobacterium tumefaciens* suspension for 20 min and then co-cultured on co-cultivation medium [41] in the dark at 28 °C for 3 days in vitro. After co-cultivation, the calli were rinsed to remove *Agrobacterium*, and then, the rinsed calli were transferred to selection medium [41] and cultured in the dark at 27 °C for 15 days in vitro. This selection process was repeated three times, with the hygromycin concentration being sequentially increased to 30, 50, and 70 mg/L. During the selection process, carbenicillin (250 mg/L) and cefotaxime (50 mg/L) were added to the selection medium to prevent *Agrobacterium* contamination. The selected resistant calli were transferred to a regeneration medium and cultured under light at 28 °C for 7 days. Once the calli turned green, developed shoots, and formed roots, the plantlets were moved to a rooting medium and maintained under light at 28 °C for 5 days. Then, the seedlings were transplanted into pots in a greenhouse (under the above culture conditions).

For obtaining transgenic rice, DNA from transformants was used as the template to amplify the *Hyg* gene and target gene, *TpGSDMT*, by PCR. The primers used for *Hyg* are 5′-GAGCATATACGCCCGGAGTC-3′ and 5′-CAAGACCTGCCTCTGAAACCGA-3′. The primers used for *TpGSDMT* are 5′-ATGGCGCCTAATACCAGTACATCAACG-3′and 5′-TTAGCTGGATGGTGATTCGTCTACAACC-3′. Seeds of each T1 generation (obtained from the positive transgenic plants mentioned above) were germinated in water containing 50 mg/L hygromycin. Seeds that germinated successfully, exhibited robust growth, and from which both *Hyg* and *TpGSDMT* could be amplified simultaneously were identified as positive transgenic rice. Then the positive transgenic lines were subsequently transplanted into the field (see above). After four months, seeds (T2 generation) were individually harvested from each transgenic rice plant. Due to the expected segregation of traits in the T2 generation, 100 seeds from each T2 line were subjected to the same germination assay in hygromycin-containing (50 mg/L) water. If all 100 seeds germinated and showed healthy growth and both *Hyg* and *TpGSDMT* could be amplified simultaneously, the corresponding T2 line was confirmed to be homozygous. In case a resistant-to-sensitive segregation ratio of approximately 3:1 was observed, it indicated that the corresponding T2 line was heterozygous. Furthermore, if all seeds failed to germinate or exhibit poor growth, it indicated that the corresponding rice was non-transgenic.

For *TpGSDMT* expression analysis, WT and homozygous transgenic T2 seeds (three independent lines; the screening method was performed as described in Section 2.2) of *TpGSDMT* overexpression were surface-sterilized and incubated on 1/2 MS basal medium plates (40 per plate) under standard conditions (28/20 °C, day/night) with supplemental light (16 h photoperiod); after 7 days, the leaves of T2 transgenic rice and WT were harvested, and RNA was isolated using an EasyPure Plant RNA Kit (TransGen Biotech, Beijing, China). The first strand of the reverse-transcribed cDNA was synthesized using a Monad first-strand cDNA Synthesis Kit (Vazyme, Nanjing, China). For RT-qPCR, the rice *ACTIN* gene (NM_001423082.1, LOC4349087) was selected as a reference gene. Forward and reverse primers for RT-qPCR analysis were designed for *OsACTIN* (forward, 5′-AGCTATCGTCCACAGGAA-3′; reverse, 5′-ACCGGAGCTAATCAGAGT-3′) and *TpGSDMT* (forward, 5′-GGGCGTCTATCTCTGGCTA-3′; reverse, 5′-CTCCAACAAATCATGGCGGC-3′). The Tb Green^®^ Premix Ex Taq™ II (Takara, Beijing, China) was used for RT-qPCR; all data were obtained from three biological repetitions, and the experiments were repeated three times. The RT-qPCR conditions were as follows: 95 °C for 30 s, followed by 40 cycles of 95 °C for 5 s and 60 °C for 20 s. The relative expression level was determined by the 2^−ΔΔCt^ method.

### 2.3. GB Extraction and Quantification

GB was extracted as described by Yu et al. with some modifications [41]. Approximately 1.0 g of fresh leaves of homozygous transgenic T2 plants (three independent lines; the screening method is the same as above) or WT rice plants obtained as detailed in Section 2.2 (see above) was ground in liquid nitrogen and transferred into a centrifuge tube. A volume of 20 mL of deionized water was added to the tube and then extracted in a water bath at 80 °C for 1 h. After centrifugation at 5000× *g* for 15 min at 4 °C, the supernatant was collected and filtered through a 0.22 μm filter. The filtrate was frozen at −80 °C overnight and then freeze-dried using a vacuum freeze dryer. The dried sample was dissolved in 1 mL of deionized water, and the solution was purified by an anionic resin (Dowex AG1 OH^−^, 200–400 mesh, Dow Chemical Co., Midland, MI, USA). GB content was analyzed using high-performance liquid chromatography (HPLC) with an Agilent 1200 (Agilent Technologies, Inc., Santa Clara, CA, USA). A ZORBAX-NH2 HPLC column (250 mm × 4.6 mm, 5 m; Agilent) was maintained at 28 °C in the column oven. The flow rate of the mobile phase (85% acetonitrile; 15% H_2_O) was 1 mL min^−1^, and the detection wavelength was 195 nm. Data were analyzed using Agilent ChemStation software (B.04.00), and the amount of GB in each sample was estimated from peak surface areas by referring to standard GB solutions.

### 2.4. Analysis of Tolerance to Salt and Low-Temperature Stresses of Transgenic Rice

In order to assess salt stress tolerance, the uniform seedlings of WT and transgenic T2 plants (three independent lines) obtained as detailed in Section 2.2 (see above) were selected and transferred to tissue culture bottles containing different concentrations of NaCl (0, 50, 100, and 150 mM). The incubation condition was described above. After 3 days, primary-root length, shoot height, and fresh weight were measured. Three independent experiments, each with 10 plants per line, were performed in WT and transgenic plants. To evaluate low-temperature stress tolerance, the uniform seedlings of WT and transgenic plants were selected and transferred to tissue culture bottles, and the bottles were placed in a growth chamber at 10 °C with a 16 h/8 h light/dark cycle.

### 2.5. Statistical Analysis

The data are presented as means ± SE. Mean comparison was performed using one-way analysis of variance (ANOVA) or Student’s *t*-tests in SPSS Statistics 27.0 (SPSS Inc., Chicago, IL, USA), and statistical significance was considered at *p* < 0.05 or *p* < 0.01. All figures were created using OriginPro 2023.

## 3. Results

### 3.1. Identification of Transgenic Rice

A binary vector containing the *TpGSDMT* gene driven by the CaMV 35S promoter was constructed (Figure 1) and transferred into rice via *Agrobacterium*-mediated transformation. A total of 32 rice transformants were finally obtained. Primers specific to the hygromycin phosphotransferase (Hyg) antibiotic resistance gene and the *TpGSDMT* target gene were used to select and confirm the presence of the transgene in transformed rice by PCR (Figure 2). A total of 21 transgenic plants were obtained, with a transformation efficiency of 65.6%. Three independent homozygous T2 transformant lines (L5, L14, and L22) were selected for further analyses (the screening method was performed as described in Section 2.2).

### 3.2. Expression Analysis of TpGSDMT in Transgenic Rice

In order to determine the expression level of *TpGSDMT*, real-time fluorescence quantitative PCR (RT-qPCR) was used. The results show that the transcript levels of *TpGSDMT* varied among the L5, L14, and L22 lines of transgenic T2 plants; however, there was no expression in the WT plants (Figure 3). The highest expression level of *TpGSDMT* was observed in L22, followed by L14 and L5. Moreover, the results indicate that the *TpGSDMT* gene was overexpressed in these transgenic lines.

### 3.3. Determination of GB Content in Transgenic Rice

To determine the level of GB in three transgenic plant lines (L5, L14, and L22), quantification of GB was performed using high-performance liquid chromatography (HPLC). As shown in Figure 4, the content of GB in the leaves of transgenic lines L5, L14, and L22 ranged from 4.8 to 5.8 μmol·g^−1^; FW, which corresponded to the transcript levels of *TpGSDMT* and was significantly higher (*p* < 0.05) than that in WT plants (where no GB was detectable) (Figure 4). These results demonstrate that *TpGSDMT* was successfully expressed and significantly enhanced GB synthesis in the transgenic rice plants.

### 3.4. Salt and Low-Temperature Tolerance of Transgenic Rice

Since the overexpression of *TpGSDMT* transgenic lines accumulated high levels of GB, we further tested the resistance of transgenic lines to salt and low temperature. WT and T2 transgenic seeds (L5, L14, and L22) of similar size and uniform weight (Appendix A) were selected for these experiments. The results show that the transgenic lines were more resistant to salt stress (50, 100, and 150 mM NaCl) or low temperature (10 °C) than the WT plants (Figure 5a and Figure 6a). Furthermore, we determined the root length, shoot height, and fresh weight of these seedlings. The results showed that under different salt concentrations or 10 °C treatments, the root length (Figure 5b and Figure 6b), shoot height (Figure 5c and Figure 6c), and fresh weight (Figure 5d and Figure 6d) of transgenic rice seedlings were significantly (*p* < 0.05 or *p* < 0.01) higher than those of WT plants, revealing that they were more resistant to salt and low-temperature stresses in the *TpGSDMT*-overexpressing transgenic rice lines.

## 4. Discussion

In the present study, *TpGSDMT*, which catalyzes successive three-step methylation of glycine to form GB, was isolated from *Talassiosira pseudonana* and transferred into rice, a GB non-accumulator. In *TpGSDMT* transgenic rice, *TpGSDMT* had a large number of transcripts; in contrast, WT rice plants, a species lacking the glycine methylation pathway, showed no detectable expression of *TpGSDMT* (Figure 3). The present data clearly indicate that GB biosynthesis was significantly enhanced in *TpGSDMT* transgenic rice plants, and it was closely correlated with the *TpGSDMT* expression level (Figure 3 and Figure 4). Nevertheless, GB was almost undetectable in WT rice (Figure 4). Meanwhile, under salt and low-temperature stresses, *TpGSDMT* transgenic rice exhibited significantly improved stress resistance compared with WT rice (Figure 5 and Figure 6). Previous studies have shown that abiotic stresses such as salt and low temperature can induce excessive ROS accumulation in mesophyll cells, root tips, stem apex, etc.; this subsequently impairs the activity of electron transport proteins in photosystem II and interferes with the synthesis, polar transport, and distribution of auxin in roots [55,56,57,58]. Ultimately, these disturbances lead to hormonal imbalance in different cell types and disrupt growth kinetics, resulting in the inhibition of plant growth. Furthermore, salinity stress can trigger critical destabilization of the symmetric distribution of cellular contents, disrupt cell division, and ultimately result in cell death [59,60]. Additionally, abiotic stresses impair water uptake, leading to increased electrolyte leakage, MDA accumulation, and reduced relative water content in plants [61]. In our study, the overexpression of *TpGSDMT* significantly increased GB levels in transgenic rice, thereby enhancing its tolerance to both salt stress and low temperature (Figure 4, Figure 5 and Figure 6). We hypothesize that the accumulation of GB in rice seedlings alleviates the imbalance between cell types and organelles caused by salt stress and low temperature. This may modulate ROS levels and facilitate adaptation through intercellular communication. In addition, our results demonstrated that the transgenic rice exhibited longer roots and shoots compared with the WT (Figure 5 and Figure 6), suggesting that the transgenic lines have resistance to the inhibition of hormonal gradients. However, the specific mechanism underlying the relationship between GB, hormonal gradient, and root and shoot growth under stress conditions requires further investigation and represents a key focus of our future research.

Strikingly, the GB content (4.8–5.8 μmol g^−1^ FW; Figure 4) in *TpGSDMT* transgenic rice was higher than that achieved by previous genetic engineering for GB biosynthesis in non-GB-accumulating plants. For example, in *CodA* transgenic tomato, GB content was about 1.5–2.4 μmol g^−1^ FW [62], and in *OsCMO* transgenic tobacco, GB content was only 0.18–0.47 μmol g^−1^ FW [40]. This result can be explained by two factors. Firstly, both *CodA* and *CMO* use choline as the substrate to synthesize GB, but the content of endogenous choline in non-GB-accumulating plants is insufficient [40,41]. Therefore, the desirable GB content cannot be achieved by transferring the genes that use choline as a substrate into non-GB-accumulating plants. And secondly, the substrate of *TpGSDMT* is glycine, which is the major amino acid that can be synthesized in all living organisms; furthermore, the high expression of SAM (S-adenosylmethionine) synthetase in plants has also been reported [42]. Consequently, *TpGSDMT* transgenic rice has the necessary substrate and methyl donor to synthesize GB.

However, in *ApGSMT*/*ApDMT* transgenic *Arabidopsis* and *ApGSMT*/*ApDMT* transgenic rice, these transformants can also synthesize GB using glycine as the substrate, but the GB content in these transformants (0.2–2.4 μmol g^−1^ FW) was much lower than in *TpGSDMT* transgenic rice (4.8–5.8 μmol g^−1^ FW) [42,43]. One reason for this result might be that the methyltransferase activity of *TpGSDMT* was higher than that of *ApGSMT* or *ApDMT*, and this hypothesis will also be considered in our further study program. Another possibility is that the synergistic action of two genes, *ApGSMT* and *ApDMT*, is required to catalyze the three-step methylation of glycine to form GB in *ApGSMT*/*ApDMT* transgenic *Arabidopsis* or rice [42,43]. But in *TpGSDMT* transgenic rice, the three-step methylation of glycine only requires *TpGSDMT*. Notably, although GB content in *TpGSDMT* transgenic rice was much higher than in other transgenic plants, it was still much lower than the levels (124–360 μmol g^−1^ FW) measured in GB-accumulating plant species such as spinach, wheat, and corn [40,41,43,44]. It seems that with the genetic modification of the GB synthesis pathway alone in non-GB-accumulating plants, it is difficult to achieve the GB level of GB-accumulating plants. After all, the synthesis of GB in plants is affected by many factors, including ROS-scavenging enzymes, Na^+^/H^+^ antiporters, heat-shock proteins, and chaperones [63,64,65]. Substantial work remains to be conducted to increase the content of GB in non-GB-accumulating plants. However, up to now, our study represents the simplest and most efficient way to increase GB content in non-GB-accumulating plants. And our results provide excellent genetic resources for molecular rice breeding.

## 5. Conclusions

Abiotic stresses such as salinity and low temperature are major limiting factors for crop cultivation, while GB plays a vital role in abiotic stress tolerance. Rice, a major cereal crop that is typically unable to accumulate GB, is particularly sensitive to these stresses. Therefore, conferring the ability to synthesize GB to rice through genetic engineering is very important. In view of this, we isolated the *TpGSDMT* gene, which catalyzes three successive methylation steps of glycine to form GB from *Talassiosira pseudonana*, and transferred it into rice. In *TpGSDMT* transgenic rice, *TpGSDMT* had a large number of transcripts, and the GB content was significantly enhanced in *TpGSDMT* transgenic rice plants and was closely correlated with the *TpGSDMT* expression level. Meanwhile, *TpGSDMT* transgenic rice had greater resistance to salt and low-temperature stresses than WT rice. Notably, as of now, our research study represents the simplest (only *TpGSDMT* needs to be transferred into rice) and most efficient (GB content of up to 5.8 μmol g^−1^ FW) way to increase GB content in non-GB-accumulating plants. This study provides considerable promise for the molecular design breeding of abiotic stress-tolerant rice.

## Figures and Tables

**Figure 1 biomolecules-15-01576-f001:**
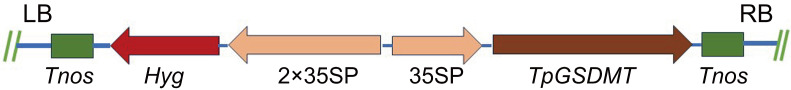
Schematic diagram of part of the T-DNA region of the construct of pBWA(V)HS-*TpGSDMT* used for *Agrobacterium*-mediated transformation in rice. *TpGSDMT*, glycine sarcosine and dimethylglycine methyltransferase; *Hyg*, hygromycin phosphotransferase, hygromycin resistance gene used for selection of transformed rice; 35SP, cauliflower mosaic virus 35S promoter; *Tnos*, nopaline synthase polyadenylation terminator; LB, left T-DNA border; RB, right T-DNA border.

**Figure 2 biomolecules-15-01576-f002:**
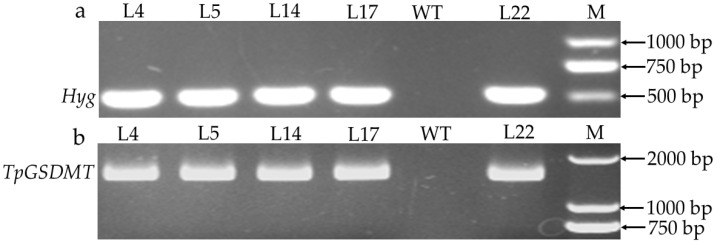
Identification of transgenic rice. Putative transgenic plants were identified by the presence of the selective gene, *Hyg* (**a**), and target gene, *TpGSDMT* (**b**), using specific primers for PCR. Five positive transgenic rice lines (L4, L5, L14, L17, and L22) were detected. Wild-type (WT) was used as negative control. M, GoldBand DL2000 DNA marker (Yeasen Biotechnology, Shanghai, China). Original PCR images are included in the Appendix A.

**Figure 3 biomolecules-15-01576-f003:**
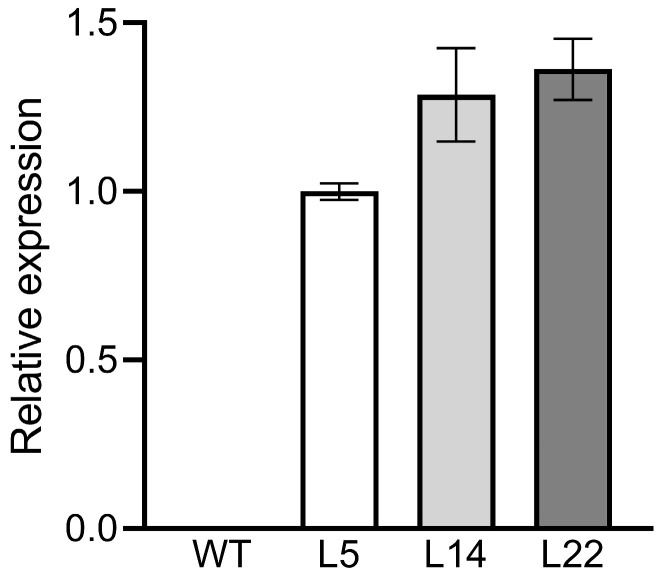
Expression levels of *TpGSDMT* in WT and transgenic rice. The leaves of T2 transgenic lines L5, L14, and L22 were used to analyze the expression level of *TpGSDMT* by RT-qPCR. Data are presented as means ± standard error (SE) (*n* = 3).

**Figure 4 biomolecules-15-01576-f004:**
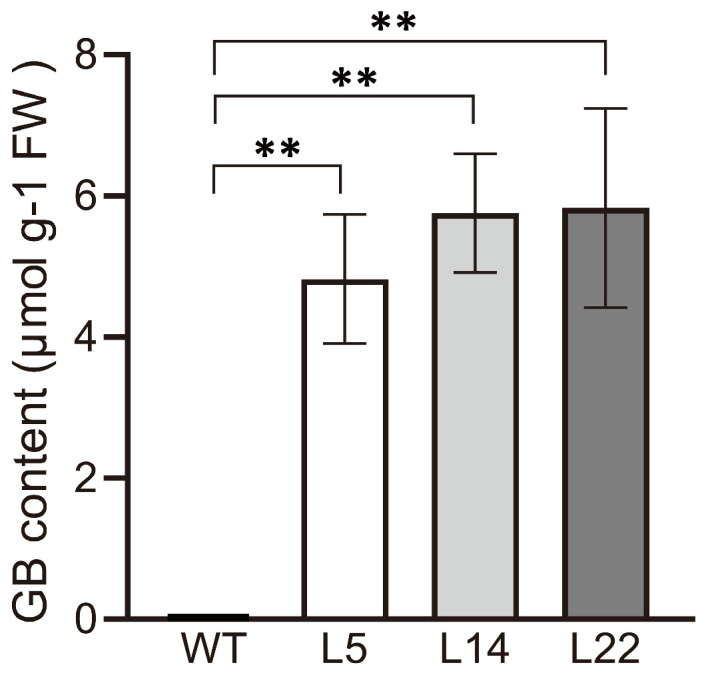
GB concentration in WT and transgenic T2 lines (L5, L14, and L22). Leaf samples from WT and three independent transgenic lines were analyzed by HPLC. Data are presented as means ± SE (*n* = 3). Asterisks indicate significant differences (** *p* < 0.01) compared with the WT using Student’s *t*-test. FW, fresh weight.

**Figure 5 biomolecules-15-01576-f005:**
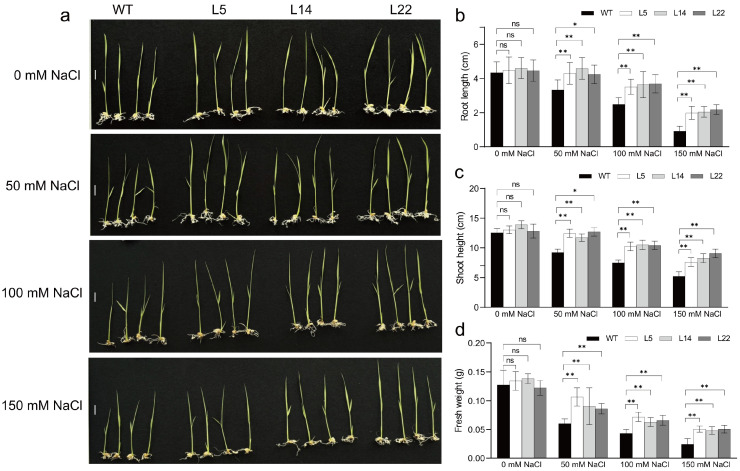
Seedling phenotypes (**a**) and root length (**b**), shoot height (**c**), and fresh weight (**d**) of WT and transgenic lines (L5, L14, and L22) under salt stress. Transgenic T2 seeds of each line were germinated in 1/2 MS, and after 4 days, 10 uniform seedlings of each line were transferred to 1/2 MS tissue culture bottles containing different concentrations of NaCl (0, 50, 100, and 150 mM). The root length, shoot height, and fresh weight were measured 3 days later. Data are presented as means ± SE (*n* = 3). Asterisks indicate significant differences (* *p* < 0.05 or ** *p* < 0.01) compared with the WT using Student’s *t*-test. ns, no significant. Bar = 1 cm.

**Figure 6 biomolecules-15-01576-f006:**
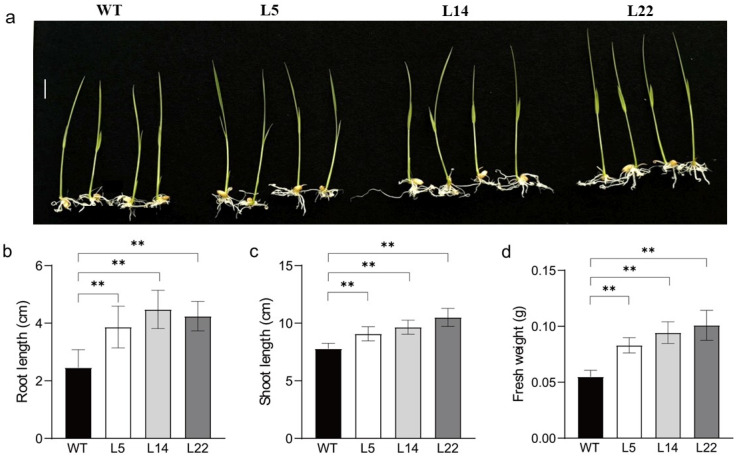
Seedling phenotypes (**a**), root length (**b**), shoot height (**c**), and fresh weight (**d**) of WT and transgenic lines (L5, L14, and L22) under low-temperature (10 °C) stress. Transgenic T2 seeds of each line were germinated in 1/2 MS, and after 4 days, 10 uniform seedlings of each line were transferred to 1/2 MS tissue culture bottles, which were placed in a growth chamber at 10 °C. The root length, shoot height, and fresh weight were measured 3 days later. Data are presented as means ± SE (*n* = 3). Asterisks indicate significant differences (** *p* < 0.01) compared with the WT using Student’s *t*-test. Bar = 1 cm.

## Data Availability

All relevant data are within the manuscript.

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
