# Peer review of "Overexpression of TpGSDMT in Rice Seedlings Promotes High Levels of Glycine Betaine and Enhances Tolerance to Salt and Low Temperature"

_biomolecules, 2025, doi:10.3390/biom15111576_

Round 1
Reviewer 1 Report
Comments and Suggestions for Authors
The manuscript Overexpression of TpGSDMT promotes high levels of glycine betaine and enhances the tolerance to salt and low-temperature in rice seedlings by authors Jinde Yu, Zihan Zhang, Ning Zhao, Xiaofei Feng, Dan Zong, Lihua Zhao considers the technology of improving the stability in the regulation of osmotic pressure in plastids and cytosol by enzymatic conversion of choline to glycine betaine.
This technology has proven itself well and the methodological approach proposed by the authors is justified.
The manuscript is designed according to the rules and contains all the necessary sections.
However, the physiological part is expressed rather weakly. It is well known that low concentrations of salts have a stimulating effect on plant growth, and an increase in osmotic pressure inside is usually accompanied by a slight inhibition or does not affect growth if other factors are observed. However, the authors observe some increase in the growth of the obtained transgenic lines, how can this be explained?
Another problem is that the authors did not reflect the methodology for obtaining transgenic rice, as well as obtaining its transgenic generations. What was the efficiency of transformation, where is the data excluding contamination, there is no understanding of the structure of the vector construct and the analysis of the marker gene, there is no explanation for why the authors used doubling of the promoter only hygromycin phosphotransferase, but did not use it for the target gene. There is no information on the methods of assessing and selecting regenerants, which is key to this technique.
There are also questions about the presented images
It is completely unclear why we do not see roots in the presented seedlings, why there is no analysis of their number, this is very strange, especially since the authors posted photos of plants with roots above, albeit of poor quality. This requires correction and explanation.
Another problem is that the authors ignore the discussion of works indicating similar experiments. We obtained tobacco plants in which glycine betaine was obtained quite effectively and the regenerants were evaluated using the salinity factor (Raldugina G. N. et al. Heterologous codA gene expression leads to mitigation of salt stress effects and modulates developmental processes //International Journal of Molecular Sciences. - 2023. - Vol. 24. - No. 18. - P. 13998). In this case, our enzyme is delivered directly to the substrate source - to the plastid, and this point is ignored in the authors' design. Although there are explanations of how this can be done, the issue is not discussed in the introduction or in the discussion. In addition to our work, there are other works where these issues are discussed in detail. I believe that they should be reflected in the introduction and discussion of physiological effects.
The methodology and quantitative data should also be expanded, including a description of the cultivation of plants and evaluation of first and second generation transgenicity, where splitting can usually be observed.
I also recommend a detailed description of the transformation procedure, the Agrobacterium strain, explants, and the methodology for selecting and evaluating transgenic plants.
After correction, the article can be reviewed.
Reviewer 2 Report
Comments and Suggestions for Authors
The article Overexpression of TpGSDMT promotes high levels of glycine betaine and enhances the tolerance to salt and low-temperature in rice seedlings by authors Jinde Yu, Zihan Zhang, Ning Zhao, Xiaofei Feng, Dan Zong, Lihua Zhao considers the issue of using glycine betaine as a modulator of plant protection from abiotic damage.
The manuscript is designed according to the rules and contains the necessary sections.
However, I find that this manuscript has a number of shortcomings both in the design and in the presentation and discussion of the concept and results of the work.
The authors have not determined the threshold concentrations of salt that prevent the germination of the original genotype. The photographs provided do not contain a ruler (Figure 5, 6) it is not possible to see significant inhibition of the roots, since the magnification and quality of the photo are critically low. As for the inhibition of the aboveground part, it is almost imperceptible in these images, which either indicates the absence of inhibition (i.e. the variety is a priori resistant), or that the photos were taken incorrectly or selected incorrectly. simple measurements demonstrate the lack of correspondence with the histograms.
The root system should be visible, at least as Ahmed, N.; Zhu, M.; Li, Q.; Wang, X.; Wan, J.; Zhang, Y. Glycine Betaine-Mediated Root Priming Improves Water Stress Tolerance in Wheat (Triticum aestivum L.). Agriculture 2021, 11, 1127. https://doi.org/10.3390/agriculture11111127Please check this.
It is known that salinity and cold, in addition to measuring growth associated with cytoskeleton disruption and growth by stretching due to decreased microfibril mobility and a drop in osmotic pressure, the authors do not provide any physiological data. What is the reason?
Also, the analysis of the provided photographs by coloristics does not demonstrate the color effects characteristic of salinity and cold. It is not clear how to explain this. Why is there no photo of the roots if they are curved, which often happens with impaired sugar transport and impaired gravitropism, then how can we explain that the roots are not visible under stress? Either improve the quality or replace the photo so that it looks correct.
One of the main questions for this publication is the aspect of obtaining and selecting transgenic plants and their generations during splitting (the authors do not provide any data). The authors do not demonstrate or describe the regeneration of generation 0, the results of splitting in the first generation. no quantitative indicators, no qualitative indicators!
Please describe in full: how the experiment was set up, how many plants and lines there were, how the cultivation and generation analysis were carried out, was there a difference in setting (osmotics are known to affect pollen and fertilization, as well as the size of seeds and fruits, although the data is contradictory), provide data on the weight of seeds in the norm and in transgenic lines, provide photos confirming their uniformity, because the weight of the grain affects the stability of developing seedlings!
There are also questions about the theoretical component of this manuscript. A number of works directly related to transgenic plants with glycine betaine are not discussed: Raldugina, G. N., Bogoutdinova, L. R., Shelepova, O. V., Kondrateva, V. V., Platonova, E. V., Nechaeva, T. L., ... & Baranova, E. N. (2023). Heterologous codA gene expression leads to mitigation of salt stress effects and modulates developmental processes. International Journal of Molecular Sciences, 24(18), 13998.
Also not discussed are the cytological aspects of growth reduction under salinity, which are reflected, for example, Shao, J., Tang, W., Huang, K., Ding, C., Wang, H., Zhang, W., ... & Qari, S. H. (2023). How does zinc improve salinity tolerance? Mechanisms and future prospects. Plants, 12(18), 3207.
I recommend that the authors not only read the proposed works, but also significantly rework the concept of the discussion and introduction to explain how exactly salinity and cold inhibit the growth of cells and organs in length.
In connection with the above, I find it important to correctly change the last paragraph of the introduction by more clearly formulating the tasks and accordingly changing the conclusion and abstract.
Also, the assertion that this method of creating plants with glycine betaine is better than others is rather dubious - how can the authors demonstrate this?
If the authors can eliminate technical errors and radically revise the results and discussion, I think the work can be considered. In its current form, I cannot recommend this publication.
Reviewer 3 Report
Comments and Suggestions for Authors
The current text devoted to inserting of glycine-betaine into rice plants.
Authors introduced TpGSDMT gene from Talassiosira pseudonana to rice plants
and showed that the transgenic rice accumulated GB strongly resistance to stress.
Ther results are significant, but the text need corrections.
Some details.
Line 9: “serious abiotic stress” ?? = significantly inhibited growth or so..
Line 23- space between words/sentences.
5 times abiotic stress in abstract. Maybe you can reduce it.
Introduction:
Comprehensive, but grammar and some messages need polishing and clarification.
Line 39: “GB activates the reactive oxygen species 39 (ROS) scavenging system” ? Directly or indirect?
Lines 39-46: there are too much different functions which require some “governor”
Line 94: “ stabilizing the condition of plants” ??
Line 99: “It has been found that GB can stimulates plant growth through a comprehensive improvement in photosynthesis and primary metabolism, which in turn enhances the salt stress response mechanisms of tomato[49]. – very long but confusing sentence, as an example. What is primary metabolism here? What is salt stress response?
Many sentences have similar problems.
Lines 143- 146 _ in vitro? medium? Conditions?
Line 161: “respectively??
Line 195: “EP tube. Adding 20 mL deionized water” – what is EP tube?
Line 211: to square plates – what is “square plates” ??
Line 230: “transgenic positive plants” = transgenic plants
Fig 3. What is relative expression here? Compare with what?
Lines 353- 356: please, provide the physiological mechanism.
Line 357 “excessive ROS production,” – or accumulation? In which location? Cell type? “damages proteins, DNA, and cel-357 lular membranas” – this is not true. Please, mention correct ROS effect – as disbalance between different cell type in hormonal balance, growth kinetics and as a results inhibition of plant growth. Plant growth is coordination between many cell types and stress, and ROS induce misbalance between it.
Line 368 “repair of plastid structure”??
Lines 353- 371: there are too much hypothesis, but your hypothesis of the mechanism – target, butterfly effect etc is missing. It will be great to describe it at least as possible hypothesis, based on plant as system.
Round 2
Reviewer 1 Report
Comments and Suggestions for Authors
The article "Overexpression of TpGSDMT promotes high levels of glycine betaine and enhances tolerance to salt and low temperatures in rice seedlings" by Jinde Yu et al. has been edited and supplemented. Overall, the work now meets the journal's requirements.
However, when reading the responses, the following point raised concerns:
After this frightening phrase, "the 35S promoter is followed by a recombination site to facilitate the insertion of the target gene," I strongly recommend consulting with the specialists who created this construct. Does this phrase imply that the recombination site is located in the expression cassette between the 35S promoter and the target gene's translation initiation site? Then, during the proposed insertion, the promoter could integrate into one region of the genome, and the target gene, without a promoter, into another region of the genome. This needs to be clearly explained. Let's assume the vector was treated with class IIS restrictases (Bsa and Eco31 are at the same site), but which restrictases were used to treat the PCR product? Insert it. This needs to be clarified.
Please carefully review the molecular section to avoid errors and make corrections according to the actual situation.
Reviewer 2 Report
Comments and Suggestions for Authors
The article "Overexpression of TpGSDMT promotes high levels of glycine betaine and enhances the tolerance to salt and low temperatures in rice seedlings" by Jinde Yu, Zihan Zhang, Ning Zhao, Xiaofei Feng, Dan Zong, and Lihua Zhao has been revised and expanded.
I recommend the authors add a clarification on why they used a dual promoter for the marker gene and a single promoter for the target gene, as this may raise questions for those who decide to repeat this work in the future.
R.S. If the histograms can be colored, it would be better if they were, as this improves readability.
Reviewer 3 Report
Comments and Suggestions for Authors
Thank you for revision! Please, before resubmission check all layout: space between words/citations is a key. Layout is a "face" of your work!
Details:
Line 23: " rice(Oryza sativa L. ssp. Japonica)" - space between worrds. Also lines 28, 90 and more..
Line 44: "enhanc the fruit" ??
Line 115 " genetic engineering strategy" for imporvement stress resistance.
It will be nice to provide at least briefly some mechanism of salt stress as imbalance between different cell type in sodium sensitivity and vacuaolr pH regulation.
Line 257: "real-time fluorescence quantitative PCR (RT-qPCR) was used to analyse. The result showed that the transcripts of TpGSDMT were expressed in different extents " = qPCR. Transcript can not expressed. Transcript level or gene exoression.
Line 362: punctuation - point at the line begin.
Line 384: "caused by salt or low-temperature-induced ROS" ??? This is not correct: salt stress induce imbalance between cell type and between orgaanelle inside cell, which can lead to ROS accumulation and therefore, adaptaion through intercelular communication. Your results with longer root and shoot in transgenic plants suggest about resistat to hormonal gradient inhibition (auxin gadient maintenance is a key in root growth, for example). Please, consider this in discussion and for the future.
Line 391: " was much higher than the genetic engineering of GB biosynthesis" ????
